# Behavioral and Psychological Symptoms of Dementia (BPSD): Clinical Characterization and Genetic Correlates in an Italian Alzheimer’s Disease Cohort

**DOI:** 10.3390/jpm10030090

**Published:** 2020-08-14

**Authors:** Catia Scassellati, Miriam Ciani, Carlo Maj, Cristina Geroldi, Orazio Zanetti, Massimo Gennarelli, Cristian Bonvicini

**Affiliations:** 1Genetics Unit, IRCCS Istituto Centro San Giovanni di Dio Fatebenefratelli, 25123 Brescia, Italy; c.scassellati@fatebenefratelli.eu (C.S.); cmaj@uni-bonn.de (C.M.); gennarelli@fatebenefratelli.eu (M.G.); 2Biological Psychiatry Unit, IRCCS Istituto Centro San Giovanni di Dio Fatebenefratelli, 25123 Brescia, Italy; 3Molecular Markers Laboratory, IRCCS Istituto Centro San Giovanni di Dio Fatebenefratelli, 25123 Brescia, Italy; mciani@fatebenefratelli.eu; 4Institute of Genomic Statistics and Bioinformatics, University of Bonn, 53127 Bonn, Germany; 5Alzheimer’s Research Unit-Memory Clinic, IRCCS Istituto Centro San Giovanni di Dio Fatebenefratelli, 25123 Brescia, Italy; cgeroldi@fatebenefratelli.eu (C.G.); ozanetti@fatebenefratelli.eu (O.Z.); 6Section of Biology and Genetic, Department of Molecular and Translational Medicine, University of Brescia, 25123 Brescia, Italy

**Keywords:** behavioral and psychological symptoms of dementia (BPSD), Alzheimer’s disease (AD), neuropsychiatry inventory scale (NPI), endophenotypes, CART analysis, *MTHFR*, *APOE*, *COMT*, genetic variants

## Abstract

Background: The occurrence of Behavioral and Psychological Symptoms of Dementia (BPSD) in Alzheimer’s Disease (AD) patients hampers the clinical management and exacerbates the burden for caregivers. The definition of the clinical distribution of BPSD symptoms, and the extent to which symptoms are genetically determined, are still open to debate. Moreover, genetic factors that underline BPSD symptoms still need to be identified. Purpose. To characterize our Italian AD cohort according to specific BPSD symptoms as well as to endophenotypes. To evaluate the associations between the considered BPSD traits and *COMT*, *MTHFR*, and *APOE* genetic variants. Methods. AD patients (*n* = 362) underwent neuropsychological examination and genotyping. BPSD were assessed with the Neuropsychiatric Inventory scale. Results. *APOE* and *MTHFR* variants were significantly associated with specific single BPSD symptoms. Furthermore, “Psychosis” and “Hyperactivity” resulted in the most severe endophenotypes, with *APOE* and *MTHFR* implicated as both single risk factors and “gene*x*gene” interactions. Conclusions. We strongly suggest the combined use of both BPSD single symptoms/endophenotypes and the “gene*x*gene” interactions as valid strategies for expanding the knowledge about the BPSD aetiopathogenetic mechanisms.

## 1. Introduction 

The core clinical criteria for Alzheimer’s Disease (AD), the most common neurodegenerative dementing illness, focus on the presence of memory disturbance or other cognitive symptoms that interfere with the ability to function at work or in usual daily activities [1].

Although cognitive symptoms are the actual hallmark of disease, patients often show also a broad range of “non-cognitive” disturbances, more commonly known with the term “Behavioral and Psychological Symptoms of Dementia (BPSD)”. Highlighting the importance of these symptoms is pivotal because BPSD represents the leading reason for loss of independence and institutionalization for AD *patients* [2,3,4]. About up to 90% of people with AD can present neuropsychiatric disturbances including agitation, aggression, irritation, disinhibition, anxiety, depression, apathy, delusions, and hallucinations at onset or later in the course of the disease [5,6].

Furthermore, BPSD can be recognized individually, but more often they occur in association. Indeed, 50% of subjects with AD show at least four neuropsychiatric symptoms simultaneously [7]. Their clusterization into distinct domains based on their frequency of co-occurrence allowed the definition of distinct behavioral endophenotypes [8]. In systematic reviews of studies that applied unbiased approaches to cluster BPSD, the following domains were identified: Affective (anxiety and depression), Disinhibition/Hyperactivity (aggression, impulsivity, and motor hyperactivity), Apathy and Psychosis (hallucinations, delusions, and paranoia) [3,8,9]. However, the debate about the definition of an appropriate clusterization in dementia is still ongoing [3]. More importantly, the studies available to date are characterized by a high heterogeneity, also because a certain number of symptoms (i.e., apathy, sleep disorders, eating disturbances) are not adequately grouped [3].

The susceptibility to BPSD is somewhat unclear as well as the molecular link between BPSD and AD. Within this complex of aetiological interplay, genetic background has been considered one of the key players involved in predisposing patients to specific behavioral and psychological manifestations in AD [10]. Multiple genes, prevalently involved in the processes of neurotransmission/neurodevelopment, have been assessed for their putative influence on BPSD risk, whose findings have been often inconsistent (for review [10]). For instance, the Catechol-O-Methyltransferase (*COMT*) gene was investigated in five published papers, four of which were conducted by the same authors [11,12,13,14], and the association found with “frontal”/cognitive and “psychosis” endophenotypes was weak. COMT protein is one of the major enzymes involved in the synaptic dopamine catabolism and, thus, has a crucial role in the prefrontal cortex. The well-known functional single nucleotide polymorphism (SNP) is characterized by a G to A transition at codon 108/158 (soluble/membrane-bound COMT) resulting in a valine-to-methionine substitution, giving rise to a significant, three-to-four-fold reduction in its enzymatic activity. The presence of valine (H allele = high activity) in the coding sequence corresponds dose-dependently with reduced prefrontal dopamine levels, leading subsequently to the upregulation of striatal dopamine activity [15].

On the other hand, the Apolipoprotein E (*APOE*) gene, the main recognized genetic risk factor for late-onset AD (LOAD) (allele ε4) [16], seems to give more consistent results, with the well-known alleles ε4 and ε2 associated with specific BPSD symptoms (for review [10]).

Thus, further new biological and molecular mechanisms could be involved in BPSD genetics architecture. For instance, *COMT* also regulates the folate pathway [17]. Folate is a cofactor in one-carbon metabolism, where it promotes the remethylation of homocysteine (Hcy). Numerous studies associate the folate deficiency and the resultant increase of Hcy levels with AD [18]. Interestingly, circulating Hcy levels in AD patients with and without BPSD were higher compared to the control subjects, and the plasma Hcy concentration in AD patients with BPSD was the highest among the three considered groups [19]. Hyperhomocysteinemia has been also correlated with psychosis and depression [20,21,22]. The Methylene tetrahydrofolate reductase (*MTHFR*) gene, whose genetic defects lead to hyperhomocysteinemia, could be a good and new candidate in the aetiopathogenetic mechanisms of BPSD. Within this gene, the two most commonly studied SNPs are the C677T (rs1801133) and the A1298C (rs1801131). The T allele of the C677T polymorphism provokes the synthesis of an athermolabile variant of the enzyme leading to a reduced enzymatic activity, which in turn produce an increase in the blood Hcy levels [23,24]. Recent findings supported that this allele was associated with an increased risk of LOAD [25,26], and correlated with a significant increase of Hcy levels [27,28]. Concerning the A1298C, the C allele results in a reduced enzymatic activity, but does not influence its thermolability [29,30]. 

Based on this rationale, in this study we analyzed an Italian cohort of AD patients in order to: 1. define the clinical distribution of BPSD symptoms through the characterization of single BPSD symptoms as well as of endophenotype clusters; 2. confirm and clarify the aetiopathogenetic effect of *APOE* and *COMT* variants in relation to both single symptoms and behavioral endophenotypes, selecting functional polymorphisms such as rs429358 (Cys130Arg) and rs7412 (Arg176Cys) and Val108/158Met, respectively [15,16]; 3. evaluate for the first time the putative involvement of *MTHFR* in single BPSD symptoms and likewise behavioral endophenotypes, selecting functional polymorphisms such as C677T (rs1801133) and the A1298C (rs1801131) [23,24,29,30]; 4. highlight for individual symptoms and endophenotypes a “single gene” involvement and/or a “gene*x*gene” interaction of *APOE*, *COMT*, and *MTHFR*.

## 2. Materials and Methods

### 2.1. Sample and Clinical Evaluation

The study was approved by the Local Ethics Committee (IRCCS Istituto Centro San Giovanni di Dio Fatebenefratelli, Brescia, Italy, n. 6/2006) and conducted in accordance with local clinical research regulations. Written informed consent was obtained from all participants.

AD subjects (*n* = 362) were recruited from the Alzheimer Unit of the IRCCS Istituto Centro San Giovanni di Dio Fatebenefratelli, Brescia, Italy. All patients were unrelated Caucasian subjects residing in Northern Italy with an Italian origin descent for at least two generations. All of them were assessed at their admittance with a complete sociodemographic and clinical data collection (cognitive, behavioral, neurological, functional, and physical abilities). A multidisciplinary clinical examination was performed and the diagnosis of probable AD was established based on criteria of the “National Institute of Neurological and Communicative Disorders” and the “Stroke-Alzheimer’s disease and Related Disorders Association” [31]. To assess cognitive decline, the Mini Mental State Examination (MMSE) test was used [32], in addition to the Cumulative Illness Rating Scale for Geriatrics (CIRS-G) [33]; function abilities were evaluated with the Basic Activity Daily Living (BADL), and the Instrumental Activity Daily Living (IADL) scales [34,35].

BPSD were assessed with the Neuropsychiatric Inventory (NPI) scale [36], a fully structured interview exploring 12 behavioral and neuropsychiatric domains (i.e., delusions, hallucinations, agitation/aggression, disphoria/depression, anxiety, apathy, irritability, euphoria, disinhibition, aberrant motor behavior, sleep behavior disturbances, besides appetite and eating abnormalities). It specifically provides an individual score for each explored cognitive domain specifically obtained by multiplying the severity of symptoms (1 = mild; 2 = moderate; 3 = severe) with their frequencies (4-point scale from 1 = occasionally to 4 = very frequently, more than once a day). The NPI then yields a domain rating of frequency times severity (range = 0–12). In agreement with a group of expert clinicians, each of the 12 behavioral/neuropsychiatric symptoms was classified in three groups, according to the “severity*frequency” score: 1—symptom-free (NPI = 0); 2—with low “severity*frequencies” score (NPI from 1 to 4); 3—with high “severity*frequencies” score (NPI from 6 to 12). Moreover, NPI provides a total score that globally evaluates the 12 domains for a complete clinical picture given by the sum of the individual scores (defining a range from 0 to 144).

The adopted exclusionary criteria for subjects were: (a) A history of schizophrenia, schizoaffective disorder, delusional disorder, mood disorder with psychotic features, major depressive disorder, substance use disorder, or mental retardation according to The Diagnostic and Statistical Manual of Mental Disorders (DSM-IV) criteria; (b) severe cerebrovascular disorders, hydrocephalus and intra-cranial mass, documented by *Computed Tomography* or *Magnetic Resonance Imaging* within the past 12 months; (c) abnormalities in serum folate and Vitamin B12, syphilis serology or altered thyroid hormone levels; (d) a history of traumatic brain injury or other neurological diseases (e.g., Parkinson’s Disease, Huntington Disease, seizure disorders); (e) current acute or poorly controlled medical problems (e.g., poorly controlled diabetes or hypertension; cancer within the past five years; clinically significant hepatic, renal, cardiac or pulmonary disorders); (f) absence of knowledgeable informant who could properly report information regarding the patient’s behavior.

### 2.2. Genetic Analysis

Genomic DNA (gDNA) was extracted from peripheral blood with the PureGene genomic DNA isolation kit (Gentra Systems, Minneapolis, MN, USA). The obtained gDNA was quantified by spectrophotometric quantification using the NanoDrop microvolume sample retention system (Thermo Fisher Scientific, Waltham, MA, USA) [37]. In order to verify the possible degradation, all samples were analyzed on a 0.8% agarose gel electrophoresis and a long *Polymerase Chain Reaction* (PCR) protocol was developed for exploring genetic variability in *MTHFR*, *COMT*, and *APOE*.

In all samples, the *MTHFR* [rs1801133 (C677T) and rs1801131 (A1298C)], *COMT* [rs4680 (Val158Met)], and *APOE* [rs429358 (Cys130Arg) and rs7412 (Arg176Cys)] polymorphisms were genotyped by using the SNaPshot assay [38]. Briefly, 100 ng of gDNA of each sample were amplified in the GeneAmp PCR System 9700 (Applied Biosystems, Foster City, CA, USA), and the PCR-amplification products were used as a template in a SNaPshot Multiplex assay performed according to the manufacturer’s instructions. Finally, samples were analyzed, and the allele peak determination was performed on an ABI 3130xl Genetic Analyzer. Electrophoresis results were analyzed using the GeneMapper ID software v4.0 (Applied Biosystems, Foster City, CA, USA). 

### 2.3. Statistical Analyses

All statistical analyses were conducted by using the SPSS version 23.0 (SPSS Inc., Chicago, IL, USA). The Hardy-Weinberg equilibrium (HWE) was calculated using an online calculator (http://www.husdyr.kvl.dk/htm/kc/popgen/genetik/applets/kitest.htm) for the presence of multiallelic genotypes. 

#### 2.3.1. Principal Component Analysis (PCA)

In the first exploratory phase, the set of symptoms from the NPI scale was factorially analyzed to identify possible underlying latent variables. The Principal Component Analysis (PCA) method and Varimax rotation were performed. The number of factors was determined on the basis of eigenvalues greater than one of the Pearson’s correlation matrix, and by sharp breaks in the size of the eigenvalues using a scree plot [39]. Symptom-factor correlations, (i.e., factor loading) being greater than 0.40 in absolute value were chosen to identify a simple factor structure (i.e., factors with non-overlapping clusters of symptoms).

#### 2.3.2. Classification and Regression Tree (CART) Analysis for Single BPSD Symptoms and Endophenotypes

The Classification and Regression Tree (CART) analysis using the exhaustive Chi-squared Automatic Interaction Detector (CHAID) algorithm was performed to explore the interaction between single BPSD symptoms as well as endophenotypes and all polymorphisms in *APOE*, *MTHFR*, and *COMT* genes in the BPSD cohort [40,41]. Specifically, we performed the analyses considering ε4 carriers (ε2ε4 + ε3ε4 + ε4ε4 genotypes) versus the others indicated as *APOE* ε4 non-carriers (ε2ε3 + ε3ε3 genotypes). *APOE* allele ε4 carriers, the carriers of both alleles in 677C/T *MTHFR*, in 1298A/C *MTHFR*, and in Val108/158Met *COMT* genes polymorphisms were used as a dominant model of inheritance. Moreover, the comparisons were performed incorporating “Free group” (NPI 0) along with “Low group” (NPI 1–4) versus “High group” (NPI 6–12). This is due to the evidence that “Low group” is present in our population with a frequency < 20%. As this study does not have a longitudinal design, we assumed causality in our CART model [https://www.epa.gov/caddis-vol4/caddis-volume-4-data-analysis-classification-and-regression-tree-cart-analysis]. 

The exhaustive CHAID data mining algorithm is a nonparametric procedure that makes no assumptions of the underlying data. In the CHAID analysis, nominal, ordinal, and continuous data can be used, where continuous predictors are split into categories with approximately an equal number of observations. CHAID creates all possible cross tabulations for each categorical predictor until the best outcome is achieved and no further splitting can be performed [42]. In the CHAID technique, we can visually see the relationships between the split variables and the associated related factor within the tree. The development of the decision, or classification tree, starts with identifying the target variable or dependent variable, which would be considered the root. CHAID analysis splits the target into two or more categories that are called the initial, or parent nodes, and then the nodes are split using statistical algorithms into child nodes.

The exhaustive CHAID data mining algorithm automatically pruning insignificant nodes in a decision tree was constructed through the IBM SPSS 23 statistical package program. It works on the basis of F test if a continuous dependent variable is used as in our study (endophenotypes), whereas, if the predictor variable has only two categories (single items), the 𝜒^2^-test for independence is performed for each pair of categories of the predictor variable in relation to the binary target variable. For non-significant outcomes, those paired categories are merged.

Minimum subject numbers for the parent and child nodes were fixed at 20 and 10 for constructing an optimal decision tree structure and improving the predictive performance of the algorithms. A ten-fold cross validation was activated in the study. 

Since the analyses were conducted by using decision tree models, it was not needed to have transformed data. 

SPSS automatically made a Bonferroni adjustment to calculate the adjusted *p*-values for the merged categories to control for the Type I error rate.

To exclude the influence of age and gender, we performed the univariate analyses (ANOVA) related to BPSD endophenotypes, including the variables age or gender. Selecting ENDOPHENOTYPES (“Psychosis”, “Hyperactivity”, “Mood”, “Frontal”) as a dependent variable, AGE as covariate, SEX as an independent variable, we did not find any significant association in separate analyses (Appendix A). The only significant association was observed in the “Mood” endophenotype (Appendix A, F = 4.7; *p* = 0.032); data not confirmed by the exhaustive CHAID data mining algorithm. Thus, given the non-significant analyses, age and sex were not included in the calculations. Medications as well as comorbidity were not included in our analyses. 

The Hosmer-Lemeshow goodness of fit test was used to confirm the suitability of the trees. The interactions were given by calculating the sum of the scores relative to “severity*frequencies” of the single BSPD symptoms components as a mean risk ± SD. 

The stepwise multiple logistic regression was used to confirm the classification trees.

## 3. Results

Socio-demographic and clinical features of our Italian AD cohort, as well as the alleles, genotypes, and carriers frequencies of the all polymorphisms in *APOE*, *MTHFR*, and *COMT* genes are shown in Table 1 and Table 2. Table 1 describes also the clinical features according to stratification for gender; whereas in Table 2 we reported the HWE results where no HWE deviation was observed. 

### 3.1. Clinical Characterization according to Single NPI Scores

Three hundred and eight patients were characterized by the NPI scale. Considering the NPI scores and the relative proposed ranges (Table 3), our population was mainly characterized by agitation, irritability, night-time behavior disturbances, and aberrant motor behavior. Within each group, we found that: 52% of AD patients showed higher severity in agitation symptomatology, 48% in irritability, 42% in night-time behavior disturbances and, finally, 39% showed higher severity in aberrant motor behavior.

Slightly less representative in our sample, there were: Apathy (28% higher severity), delusions (29% higher severity), anxiety (25% higher severity), depression (22% higher severity), and hallucinations (21% higher severity).

On the contrary, appetite and eating disturbances were referred in only a minority of patients, while no one presented disinhibition or euphoria according to their caregivers’ reports.

### 3.2. Clinical Characterization according to Behavioral Endophenotypes

The results of the exploratory factor analysis on BPSD in AD, conducted by PCA and Varimax rotation, are shown in Table 4. Using the criterion of eigenvalues greater than 1 and the scree plot, PCA allowed grouping the 12 explored BPSD symptoms in four endophenotypes. These factors explained the 56% of the total variance of data. Specifically, thanks to this approach, the following clinical categories were established in our cohort: (a) “Psychosis” characterized by delusions, hallucinations, agitation, aberrant motor behavior, and night-time behavioral symptoms; (b) “Hyperactivity” with agitation, irritability, appetite and eating disturbances; (c) “Mood” associated with high loadings on anxiety, depression, and apathy; (d) “Frontal” including disinhibition and euphoria.

The endophenotypes more representative in our population were “Psychosis” (37% higher severity), and “Hyperactivity” (39% higher severity). These were followed by the “Mood” and “Frontal” endophenotypes (Table 5).

### 3.3. APOE, MTHFR, and COMT Genetic Correlates in Endophenotype Clusterization and in Single BPSD Symptoms

All polymorphisms investigated in this study within the *APOE*, *MTHFR*, and *COMT* genes showed a prevalence comparable to that reported in https://www.alzforum.org/, considering Caucasian Populations (Table 2). Globally, the prevalence of ε2, ε3, and ε4 alleles for the *APOE* gene was estimated to be 8%, 78%, and 14%, respectively; for rs1801133 *MTHFR* gene to be 39% (allele T) and 61% (allele C); for rs1801131 *MTHFR* gene to be 34% (allele C) and 66% (allele A); for Val108/158Met *COMT* gene to be 49% (allele A) and 51% (allele G).

In order to explore the genetic correlates in both approaches, we exploited the potential of genetic analysis by using the exhaustive CHAID data mining algorithm to reveal “gene*x*gene” interaction as aetiopathogenetic mechanisms in BPSD. The Hosmer-Lemeshow goodness of fit test confirmed the suitability of the trees (Appendix A). Moreover, the stepwise multiple logistic regression validated the classification trees (Appendix A).

#### 3.3.1. “Genexgene” Interactions in “Psychosis” Endophenotype and the Relative Single BPSD Symptoms

Table 6 reports the main significant results obtained in the genetic analyses performed on single BPSD symptoms. Starting by the agitation, we found that the *APOE* ε4 allele carriers showed a high risk to develop more severe symptoms (odd ratio (OR) = 1.87, 95% CI: 1.19–2.95). Concerning the aberrant motor behavior, we evidenced similar results with *APOE* ε4 allele carriers showing higher risk to develop high severity in this symptomatology (OR = 1.91, 95% CI: 1.20–3.04). In relation to *MTHFR*, we found that the homozygotes CC (C677T) showed a high risk to develop more severe delusions symptoms (OR = 1.75, 95% CI: 1.04–2.94).

On the contrary, no genetic correlates were observed for night-time behavior disturbances.

These results were further represented as decision model trees in Appendix A.

When we considered the “Psychosis” endophenotype as whole, we confirmed the effect of *APOE* ε4 allele alone (risk mean = 20.7, 95% CI: 18.5–22.9), but also a “gene*x*gene” interaction with *MTHFR* for both polymorphisms. In particular, *APOE* ε4 non-carriers along with the homozygotes CC (C677T) showed a high risk to develop psychosis (risk mean = 20.2, 95% CI: 18.9–29.3), but also with the homozygotes CC (C677T) and with the 1298A allele carriers (risk mean = 23.1, 95% CI: 18.7–27.6) (Figure 1A).

#### 3.3.2. “Genexgene” Interactions in “Hyperactivity” Endophenotype and the Relative Single BPSD Symptoms

For irritability symptoms, no genetic correlates were observed, whereas for agitation, the results are reported above. Although in an unrepresentative sample, a risk role of *APOE* ε4 allele was found with higher severity in appetite/eating abnormalities (OR = 2.06, 95% CI: 1.14–3.71) (Table 6). These results were further represented in Appendix A as decision model trees.

When we consider the “Hyperactivity” endophenotype as a whole, the most significant result is represented by the risk role of *APOE* ε4 allele (risk mean = 13.2, 95% CI: 11.8–14.7). Moreover, an interaction with the *MTHFR* C677T CC genotype and *APOE* ε4 non-carriers was observed (risk mean = 12.1, 95% CI: 9.7–14.4) (Figure 1B).

#### 3.3.3. “Genexgene” Interactions in “Mood”/”Frontal” Endophenotypes and the Relative Single BPSD Symptoms

No significant results were found for these endophenotype as well as for the relative single BPSD symptoms.

## 4. Discussion

This study wants to deeply define the clinical distribution of BPSD symptoms in an Italian cohort of AD patients by the NPI scale. To the best of our knowledge, this is the largest BPSD cohort investigated until now. In this population, “Psychosis” and “Hyperactivity” endophenotypes as well as “agitation” as single symptoms were revealed to be associated with a high severity in symptomatology. In addition, “Mood” was associated to lower mean values, and, for last there was the “Frontal” endophenotype. As major clinical information, we suggest two combinatorial approaches to characterize a BPSD cohort: through 12 individual symptoms as well as four specific behavioral and neuropsychiatric endophenotypes. This choice allows us to overcome the limitations linked to a single methodological approach, bringing to light the interconnected aspects of the complex pathogenic mechanisms of BPSD. Moreover, we strongly propose analyzing the genetic correlates also in relation to single BPSD symptoms as well as to BPSD endophenotypes. This represents a valid strategy for expanding the knowledge about the aetiopathogenetic mechanisms of BPSD. There is more. We strongly suggest the use of both strategies also in relation to the genetic approach “single gene” involvement/“gene*x*gene” interaction, because this contributes to clarifying deeply the complex genetic architecture underlying BPSD. The main results obtained in this work are represented by the involvement of *APOE* as a “single gene” in the modulation of the severity in agitation, also reflected in ”Psychosis” as well as in “Hyperactivity” endophenotypes, and in aberrant motor behavior. Moreover, we found the involvement of *APOE* also in the “gene*x*gene” interaction with *MTHFR* (both polymorphisms) as risk factors for “Psychosis”. *MTHFR* as a “single gene” was associated to higher severity of delusions. This implicates that specific functional variants in *APOE* and *MTHFR* genes can impact on the relative proteins’ translation and thus on their activity [16,23,24,29,30], to influence specific BPSD traits. No associations or minor relevant results include the “Frontal” endophenotype (euphoria and disinhibition) and the relation between *APOE* and appetite/eating abnormalities.

Neuropsychiatric symptoms are frequent in dementia and contribute significantly for burden caregiver and illness costs. Correct identification and evaluation of these symptoms is a crucial part of the clinical approach to dementia. Despite the added value and tentative efforts to group different symptoms into clusters (to facilitate clinical/diagnostic/treatments investigations), there is not yet an established model. Cerejeira et al., 2012 [3] reviewed different studies showing the heterogeneity in the clusterization of behavioral endophenotypes, even though a certain concordance can be found. Delusions and hallucinations have been consistently grouped in “Psychosis”. A distinct “Mood” or “Affective” cluster (depression and anxiety) has been suggested, in accordance with our study. On the contrary, some studies support that apathy and depression are distinct phenomena and belong to different neuropsychiatric syndromes, whereas others group both symptoms in the same factor [3], as in our case study. The frequencies of BPSD symptoms in the AD population can change [43]. As reported in Deardorff and Grossberg, 2019 [43], apathy is the most common BPSD symptom in AD (49%), followed by depression (42%), agitation and aggression (40%), psychosis (delusions, 31% and hallucinations, 16%), and sleep disturbances (39%). Such a distribution is different as compared to our BPSD/AD cohort.

These discrepancies could be related to different factors. First of all, the distribution of BPSD symptoms is strongly linked to differences in patient populations, their ethnicity, and composition (for instance whether the patients are affected by AD or by other neurodegenerative disorders). A small sample size is an important limitation. This study represents the larger cohort available to date, that permits performing a better stratification according to BPSD symptoms, where the range of symptoms has been precisely defined, besides being crucial to detect significant genetic associations. Indeed, genome-wide association studies and studies examining copy number variations performed in AD and psychosis have detected only suggestive underpowered associations in intronic SNPs (review in [44,45]). Moreover, individual symptoms evolve differently over the course of dementia. Notwithstanding the general impression that the overall level of psychopathology increases with dementia severity, they have a tendency to wax and wane, their severity fluctuating over time. For this reason, longitudinal studies are required to achieve significant insights into the evolution of BPSD during the course of disease [3]. Finally, the choice of setting for patient recruitment is a further important selection bias: Nursing home dwellers, inpatients, or patients treated in an ambulatory setting. The most inclusive population-based studies recruit “real-life” patients, making the results much more practical, although at a price of numerous medical, environmental, and drug-related confounders [10]. In our case, we have recruited patients from the department of our hospital, who in the majority of cases, enter for a control of BPSD. Thus, for definition, they are affected by disorders with a more prevalent “agitated” symptomatology rather than “apathy”.

If age is the greatest risk factor for AD, the ε4 allele of the *APOE* is the greatest genetic risk factor for LOAD [16]. ε4 carriers showed higher severity in agitation symptoms, grouping in “Hyperactivity”/”Psychosis” endophenotypes. The single association with agitation is confirmed in different studies, including recent reviews [10,46,47]. Based on neuroimaging studies, agitation is associated with volume loss in the frontal cortex, circulate cortex, insula, amygdala, and hippocampus [48]. Neurochemically, agitation is associated with decreased cholinergic activity in the frontal and temporal cortex. Interestingly, AD patients carrying the *APOE* ε4 allele show a more profound loss in cholinergic activity in the hippocampus and the cortex, and in neuroimaging studies the presence of the *APOE* ε4 allele was associated with a greater rate of hippocampal, cortical, and whole-brain atrophy [10]. On the other hand, there are also negative studies, reporting no associations of this gene with agitation [49]. These findings, thus, further emphasize the importance of sub-grouping BPSD in distinct neuropsychiatric syndromes. In fact, we found an association of *APOE* also within “Hyperactivity”/”Psychosis” endophenotypes. We therefore support the evidence on the role played by the allele ε4 as a risk factor in these endophenotypes, along with the importance of the BPSD clusterization.

The ε4 carriers showed also higher severity in aberrant motor behavior, the finding was confirmed also in other different studies [50,51,52], where they demonstrated that the *APOE* ε4 status increased the tendency toward this symptom.

Another more significant result of this work regarding the “gene*x*gene” interactions between *APOE* and *MTHFR* (both polymorphisms) as risk factors for higher severity in the “Psychosis” endophenotype, underlies the important role played by *MTHFR* as a single gene in delusions (genotype CC of C677T). This interaction was observed also in “Hyperactivity”. What is highlighted from the CART analysis, is that also the other *APOE* ε4 non carriers can play a role, and this is performed through the interactions with the CC genotype (C677T) and/or 1298A allele carriers. Different studies supported the epistatic effect between *MTHFR* and *APOE* on cognitive performance in the elderly but also in AD [53,54,55]. Moreover, plasma homocysteine concentrations are associated with increased amyloid-β (Aβ) deposition in the brain [28]. Although this effect was supported, no studies are available to confirm this interaction also in the psychosis/hyperactivity dimension. The most probable hypotheses behind this result could be that *MTHFR* contributes to the regulation of relin [56], whose expression is decreased in the hippocampus of subjects suffering from schizophrenia and major depressive disorder (MDD) [57]. Relin binds to the receptor for APOE, and might thus similarly modulate glutamatergic neurotransmission, altered in psychosis. Moreover, a significant positive correlation of serum homocysteine levels was found with delusions [58].

In relation to *COMT* as a “single gene”, we did not confirm the results reported by Borroni et al., 2006 [12,13]. Further studies are needed to clarify the role of this gene, that seems to be less relevant as compared to the findings found for *APOE* and *MTHFR* genes, also in their interaction.

If we select in PubMed the following keywords “BPSD-gene-Alzheimer”, we found 32 results where the studies have been actively published until 2013, from 2013 to 2020 there are only seven works performed on the genetics of BPSD. To date, the classic limitations linked to genetic studies remain and there is still the necessity of: larger cohorts with appropriate assessment tools and statistical methods with correction for multiple testing. There are limitations regarding inconsistent data in genome-wide association studies with prevalently intronic SNPs detection and presence of confounders influencing the associations. In this study, appropriate statistical tools such as the exhaustive CHAID data mining algorithm were used allowing to obtain the adjusted *p*-values for multiple testing, and where age and sex did not influence the calculations. The only variable not included in our analyses was “Medications”. Medications as well as comorbidity are clinical information extremely heterogeneous in a BPSD population, including our cohort (Table 1). To include “Medications” or “Comorbidity” as variables in the statistical analyses would result in being overly complex, especially in a sample of sample sizes such as ours. Thus, we suggest that future studies should consider these variables, but starting from homogeneous groups for medications and for comorbidity, on which then to investigate the genetic correlates.

Although aware that further studies are still needed, we want to ignite the enthusiasm and encourage more research to deepen the etiopathogenetic mechanisms of BPSD in AD, starting by a well-defined BPSD population, appropriate statistical methods with multiple corrections, larger sample, and “gene*x*gene” and possibly “gene*x*environment” interactions. This permits not only to facilitate a clinical/diagnostic assessment, but also and mainly to define appropriate treatments chosen that, to date, represent the major concern for the clinicals.

## 5. Conclusions

We strongly suggest the combined use of both BPSD single symptoms/endophenotypes and the “gene*x*gene” interactions as valid strategies for expanding the knowledge about the BPSD aetiopathogenetic mechanisms.

## Figures and Tables

**Figure 1 jpm-10-00090-f001:**
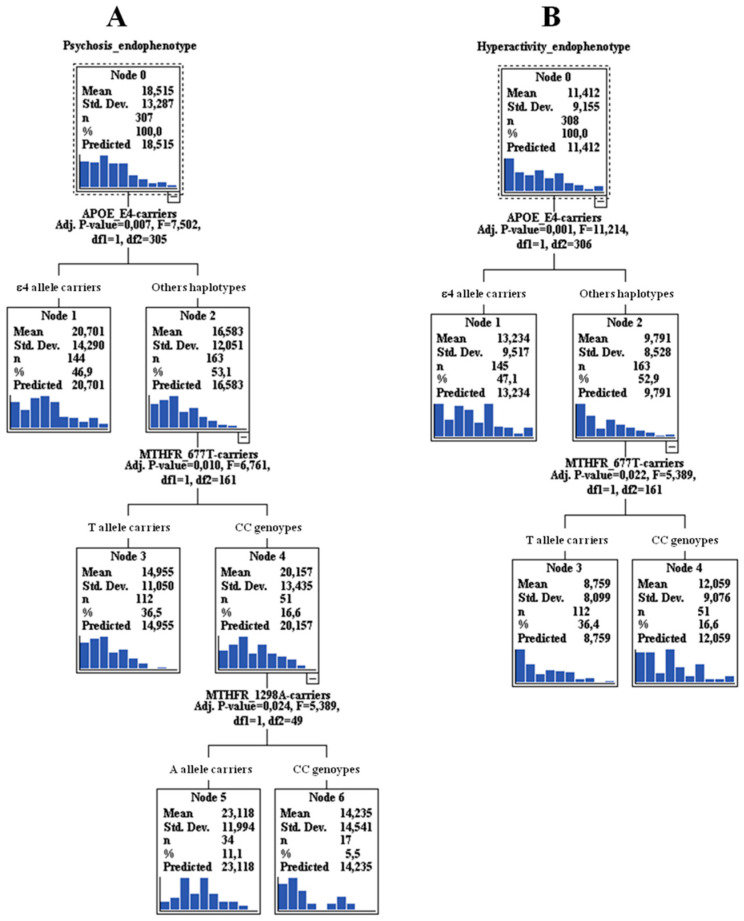
Exhaustive Chi-squared Automatic Interaction Detector (CHAID) data mining algorithm analysis results (“genexgene” interaction) for “Psychosis” (**A**) and “Hyperactivity” (**B**) endophenotypes. Note: Mean ± SD is given by the sum of the scores relative to “severity*frequencies” of the endophenotype components. For other haplotypes, we meant “*APOE* ε4 non-carriers”.

**Table 1 jpm-10-00090-t001:** Clinical features of our Alzheimer’s Disease Italian cohort, stratified according to gender.

Gender (F/M)	241/121	Female	Male
Age (years)	80.5 ± 7.0	80.9 ± 7.1	79.7 ± 6.7
Schooling (years)	6.0 ± 3.6	5.4 ± 2.9	7.4 ± 4.4
MMSE, total score	11.7 ± 7.3	12.0 ± 6.9	10.9 ± 8.1
Onset (years)	76.1 ± 7.9	76.5 ± 8.1	75.4 ± 7.6
Illness Duration (years)	4.5 ± 2.9	4.4 ± 2.6	4.5 ± 3.5
BMI	22.3 ± 4.1	22.2 ± 4.2	22.5 ± 3.8
IADL	7.2 ± 1.5	7.4 ± 1.2	6.8 ± 1.9
BADL	3.7 ± 1.9	3.6 ± 1.9	3.8 ± 1.8
Tinetti	19.8 ± 5.4	19.8 ± 5.2	19.6 ± 5.9
CIRS	12.7 ± 9.7	12.2 ± 9.5	13.8 ± 10.0
Hypertension	27.9%	29.5%	24.8%
Cardiopathy	28.7%	26.1%	33.9%
Hypercholesterolemia	4.7%	5.8%	2.5%
Diabetes	6.6%	5.4%	9.1%
Psychiatric diseases	26.0%	27.4%	23.1%
AChEIs	20.2%	21.6%	17.4%
Neuroleptics	45.5%	41.1%	51.2%
Antidepressants	45.6%	46.9%	43.0%
Benzodiazepines/hypnotics	41.4%	46.1%	32.2%

Note: The measurement data are expressed as mean ± standard deviation (mean ± SD) or in percentual (%). MMSE: Mini-Mental State Examination; BMI: Body max index; IADL: Instrumental activities daily living; BADL: Basic activities of daily living; Tinetti; CIRS: Cumulative illness rating scales; AChEIs: Acetylcholinesterase inhibitors.

**Table 2 jpm-10-00090-t002:** Genetic distribution of our Alzheimer’s Disease Italian cohort for *APOE*, *MTHFR*, and *COMT* genes polymorphisms.

*APOE* Haplotypes (rs429358 and rs7412)			*MTHFR* (rs1801133: C677T)			*MTHFR* (rs1801131: A1298C)			*COMT* (rs4680: V158M)		
HWE	χ^2^ = 0.91	*p* = 0.92	HWE	χ^2^ = 1.70	*p* = 0.19	HWE	χ^2^ = 0.53	*p* = 0.47	HWE	χ^2^ = 1.22	*p* = 0.27
*Alleles*	N	freq	*Alleles*	N	freq	*Alleles*	N	freq	*Alleles*	N	freq
ɛ2	20	0.03	T	314	0.43	A	499	0.69	A	330	0.46
ɛ3	510	0.70	C	410	0.57	C	225	0.31	G	394	0.54
ɛ4	194	0.27									
total	724	1.00	total	724	1.00	total	724	1.00	total	724	1.00
*Genotypes*	N	freq	*Genotypes*	N	freq	*Genotypes*	N	freq	*Genotypes*	N	freq
ɛ2 ɛ3	16	0.04	TT	62	0.17	AA	169	0.47	AA	70	0.19
ɛ2 ɛ4	4	0.01	TC	190	0.52	AC	161	0.44	AG	190	0.52
ɛ3 ɛ3	178	0.49	CC	110	0.30	CC	32	0.09	GG	102	0.28
ɛ3 ɛ4	138	0.38									
ɛ4 ɛ4	26	0.07									
total	362	1.00	total	362	1.00	total	362	1.00	total	362	1.00
ɛ2 carriers	20	0.06	T carriers	252	0.70	A carriers	330	0.91	A carriers	260	0.72
ɛ3 carriers	332	0.92	C carriers	300	0.83	C carriers	193	0.53	G carriers	292	0.81
ɛ4 carriers	168	0.46									

Note: *APOE*: Apolipoprotein E; *MTHFR*: Methylenetetrahydrofolate Reductase; *COMT*: Catechol-O-Methyltransferase; HWE: Hardy-Weinberg equilibrium.

**Table 3 jpm-10-00090-t003:** Clinical distributions of single Behavioral and Psychological Symptoms of Dementia (BPSD) in our Italian Alzheimer’s Disease cohort.

		“Severity*Frequencies” Groups N Subjects (%) ^&^		
Symptoms (“Severity*Frequencies” = 0–12)	Mean ± SD ^§^	Free	Low	High
		(NPI 0)	(NPI 1–4)	(NPI 6–12)
Agitation	4.96 ± 4.32	102 (33.0)	47 (15.2)	160 (51.8)
Irritability	4.57 ± 4.25	109 (35.4)	52 (16.9)	147 (47.7)
Night-time behavior disturbances	4.22 ± 4.55	136 (44.0)	44 (14.2)	129 (41.7)
Aberrant motor behavior	4.12 ± 4.72	148 (48.2)	39 (12.7)	120 (39.1)
Apathy	3.03 ± 4.19	178 (58.0)	42 (13.7)	87 (28.3)
Delusions	3.00 ± 4.00	166 (54.1)	52 (16.9)	89 (29.0)
Anxiety	2.53 ± 4.10	200 (65.4)	30 (9.8)	76 (24.8)
Depression	2.50 ± 3.90	185 (60.3)	53 (17.3)	69 (22.5)
Hallucination	2.30 ± 3.55	187 (60.9)	55 (17.9)	65 (21.2)
Appetite and eating disturbances	1.94 ± 3.59	223 (72.2)	28 (9.1)	58 (18.8)
Disinhibition	0.93 ± 2.44	253 (82.4)	28 (9.1)	26 (8.5)
Euphoria	0.12 ± 0.86	299 (97.4)	6 (2.0)	2 (0.7)
NPI, total score (0–144)	34.1 ± 22.6			
NPI, n symptoms	4.9 ± 2.4			

Note: ^§^ Mean ± standard deviation (SD) of the scores relative to “severity*frequencies” of the single items. ^&^ Number (N) and percentage (%) of subjects present in each group (Free, Low, and High) compared to individual symptoms. Free: “severity*frequencies” score (NPI 0) for all the individual symptoms; Low: “severity*frequencies” score (NPI 1–4) for all the individual symptoms; High: “severity*frequencies” score (NPI 6–12) for all the individual symptoms.

**Table 4 jpm-10-00090-t004:** Principal component analysis (PCA) of the Behavioral and Psychological Symptoms of Dementia (BPSD) in our Italian Alzheimer’s Disease cohort.

Symptoms	Endophenotype Components			
	Factor 1	Factor 2	Factor 3	Factor 4
	Mood	Hyperactivity	Psychosis	Frontal
Delusions	0.276	0.372	*0.463*	0.144
Hallucination	0.022	0.011	*0.690*	0.248
Agitation	0.058	*0.654*	*0.468*	0.068
Depression	*0.855*	0.058	0.166	0.089
Anxiety	*0.820*	0.062	0.156	0.075
Euphoria	−0.040	−0.109	0.186	*0.766*
Apathy	*0.579*	0.292	−0.212	−0.184
Disinhibition	0.131	0.376	−0.166	*0.687*
Irritability	0.028	*0.645*	0.367	0.119
Aberrant motor behavior	0.104	0.373	*0.402*	−0.157
Night-time behavior disturbances	0.029	0.056	*0.621*	−0.087
Appetite and eating disturbances	0.191	*0.671*	−0.157	0.014

Note: Extraction method: Principal component analysis. Rotation Method: Varimax with Kaiser normalization. Coefficients with values over 0.4 are italicized and in bold.

**Table 5 jpm-10-00090-t005:** Clinical distributions of behavioral endophenotypes in our Italian Alzheimer’s Disease cohort.

		Mean of N Subjects ± SD ^£^ (% Mean of N Subjects ± SD) ^$^		
Endophenotypes (Components; “Severity*Frequencies” Range)	Mean ± SD ^§^	Free	Low	High
**Psychosis** (Delusions, Hallucination, Agitation, Aberrant motor behavior, Night-time behavior disturbances; 0–60)	18.5 ± 13.3	148 ± 32 (48.0 ± 10.5)	47 ± 6 (15.4 ± 2.1)	113 ± 37 (36.6 ± 11.8)
**Hyperactivity** (Agitation, Irritability, Appetite and eating disturbances; 0–36)	11.4 ± 9.2	145 ± 68 (46.9 ± 22.0)	42 ± 13 (13.7 ± 4.1)	122 ± 56 (39.4 ± 18.0)
**Mood** (Depression, Anxiety, Apathy; 0–36)	8.1 ± 9.4	188 ± 11 (61.2 ± 3.8)	42 ± 12 (13.6 ± 3.7)	77 ± 9 (25.0 ± 3.0)
**Frontal** (Euphoria, Disinhibition; 0–24)	1.0 ± 2.7	276 ± 33 (89.9 ± 10.6)	17 ± 16 (5.5 ± 5.1)	14 ± 17 (4.6 ± 5.5)

Note: ^§^ Mean ± standard deviation (SD) of the scores relative to “severity*frequencies” of the endophenotype components. Free: NPI = 0 for all the individual components involved in the endophenotype; Low: Sum of the “severity*frequencies” score (NPI 1–4) for all the individual components involved in the endophenotype; High: Sum of the “severity*frequencies” score (NPI 6–12) for all the individual components involved in the endophenotype. ^£^ Mean number of subjects N ± SD present in each group (Free, Low, and High) compared to individual endophenotypes. ^$^ Percentage (%) of the mean of the number of subjects (N ± SD) present in each group (Free, Low, and High) compared to individual endophenotypes.

**Table 6 jpm-10-00090-t006:** Genetic correlates related to single BPSD symptoms in our Italian Alzheimer’s Disease cohort according to different values of “severity*frequencies” (exhaustive CHAID data mining algorithm).

	Gene	Free + Low ^&^	High ^&^	
		N (Freq.)	N (Freq.)	
**Agitation**	*APOE*			χ^2^ = 7.435, *p*_adjusted_ = 0.006 OR = 1.87; 95% CI: 1.19–2.95
	Ɛ4-carriers	58 (0.39)	86 (0.54)	
	Ɛ4-non-carriers	91 (0.61)	72(0.46)	
	total	149 (1.00)	158 (1.00)	
**Aberrant motor behavior**	*APOE*			χ^2^ = 7.553, *p*_adjusted_ = 0.006 OR = 1.91; 95% CI: 1.20–3.04
	4-carriers	76 (0.41)	68 (0.57)	
	Ɛ4-non-carriers	111 (0.59)	52 (0.43)	
	total	187 (1.00)	120 (1.00)	
**Delusions**	*MTHFR*			χ^2^ = 4.363, *p*_adjusted_ = 0.037 OR = 1.75; 95% CI: 1.04–2.94
	CC	59 (0.27)	35 (0.39)	
	Genotypes_C677T	159 (0.73)	54 (0.61)	
	T-carriers_C677T	218 (1.00)	89 (1.00)	
**Appetite/eating abnormalities**	*APOE*			χ^2^ = 5.922, p_adjusted_ = 0.015 OR = 2.06; 95% CI: 1.14–3.71
	Ɛ4-carriers	109 (0.44)	35 (0.60)	
	Ɛ4-non-carriers	141 (0.56)	22 (0.40)	
	total	250 (1.00)	57 (1.00)	

Note: ^&^ Free + Low: “severity*frequencies” scores (NPI 0–4) for all individual symptoms; High: “severity*frequencies” scores (NPI 6–12) for all the individual symptoms. OR: Odd ratio.

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
