# Peer review of "Behavioral and Psychological Symptoms of Dementia (BPSD): Clinical Characterization and Genetic Correlates in an Italian Alzheimer’s Disease Cohort"

_jpm, 2020, doi:10.3390/jpm10030090_

Round 1

Reviewer 1 Report

Scassellati et al have conducted a candidate gene study to determine whether several SNPs located within the genes MTHFR, COMT and APOE are associated with either individual symptoms from the neuropsychiatric inventory questionnaire or with four clusters defined using PCA.

  1. As this is a candidate gene study, the authors need to provide a robust rationale for investigating the association of these variants BPSD symptoms. Querying these variants in various GWAS databases (GWAS atlas, PhenoScanner, etc), none of these variants have been associated neuropsychiatric traits at genome-wide significance. Furthermore, COMT and MTHFR have not been associated with AD at genome wide significance. This would suggest that the associations reported by the authors are unlikely to replicate in larger studies, especially at genome wide significance. The authors should address these limitations in both in the introduction and discussion.
  2. The authors evaluated Hardy-Weinberg equilibrium, but do not report the results of this analysis.
  3. Why did the authors collapse e2/e4 carriers into the e2 carrier group, considering that e2/e4 carriers are associated with increased risk in comparison to e3/e3 carriers.
  4. The authors do not describe how their single gene analysis was conducted in the methods section, and it is not overly clear if the results from this analysis come from logistic regression model or the CART. Did the authors include any covariates in the model such as age or sex, what model of inheritance was used? The authors should report the association of each SNP with all the NPI symptoms and the four factors developed, even if they are non-significant. This should be reported in a supplementary table if need.
  5. In the CART analysis, the authors make reference to a influenza group – I presume this is a copy/paste error.
  6. How was multiple testing accounted for?

Author Response

Reviewer #1: in yellow (gray for Referees’ #1, #2 common comments)

REPLAY 1: As this is a candidate gene study, the authors need to provide a robust rationale for investigating the association of these variants BPSD symptoms. Querying these variants in various GWAS databases (GWAS atlas, PhenoScanner, etc), none of these variants have been associated neuropsychiatric traits at genome-wide significance. Furthermore, COMT and MTHFR have not been associated with AD at genome wide significance. This would suggest that the associations reported by the authors are unlikely to replicate in larger studies, especially at genome wide significance. The authors should address these limitations in both in the introduction and discussion.

RESPONSE:

We thank the Referee for these comments.

We selected the following single nucleotide polymorphisms (SNPs): [rs429358 (Cys130Arg) and rs7412 (Arg176Cys)] in APOE; Val108/158Met in COMT; and C677T (rs1801133), A1298C (rs1801131) in MTHFR genes, because these variants have been already validated and their crucial functional impact on protein translation and, in turn, activity has been widely demonstrated.

In the original version, we already reported that:

  1. “The well known functional single nucleotide polymorphism (SNP) is characterized by a G to A transition at codon 108/158 (soluble/membrane-bound COMT) resulting in a valine-to-methionine substitution, giving rise to a significant, three-to-four fold reduction in its enzymatic activity. The presence of valine (H allele = high activity) in the coding sequence corresponds dose-dependently with reduced prefrontal dopamine levels, leading subsequently to the upregulation of striatal dopamine activity [15].” (page 2 lines 71-77);
  2. “The T allele of the C677T polymorphism provokes the synthesis of an athermolabile variant of the enzyme leading to a reduced enzymatic activity, with in turn produce an increase in the blood Hcy levels [23, 24]. Recent findings supported that this allele was associated with an increased risk of LOAD [25, 26], and correlated with a significant increase of Hcy levels [27, 28]. Concerning the A1298C, the C allele results in a reduced enzymatic activity, but not influence its thermolability [29, 30]” (page 2 lines 91-96).

Notoriously, in GWAS approaches, most of detected SNPs are intronic, with limited or undemonstrated functional effects. We agree that larger sample are needed to reveal positive associations for functional SNPs, taken into account the presence of rare alleles. However, we observed significant associations in our BPSD population, and this reinforces our results and the choice to analyse the polymorphisms selected.

In the original version, we also reported that our BPSD sample is the largest studied to date, whereas “genome-wide association studies and studies examining copy number variations performed to date in AD and psychosis have detected only suggestive underpowered associations in intronic SNPs” (review in [44, 45]).” (page 12 lines 416-418).

Anyway, following the referee’s comment, we added and stressed that the rationale on which we have selected the polymorphisms was based on consistent evidences demonstrating their functional impact (introduction (page 3 lines 101-104) and discussion (page 12 lines 390-392; page 13 lines 470-471) sections).  

REPLAY 2: The authors evaluated Hardy-Weinberg equilibrium, but do not report the results of this analysis.

RESPONSE:

We thank the Referee for this point.

We inserted the results regarding the HWE in Table 2 (page 6) and in the results section (page 5, line 244). Moreover, we specified that “HWE was calculated by using online calculator (http://www.husdyr.kvl.dk/htm/kc/popgen/genetik/applets/kitest.htm), for the presence of multiallelic genotypes” (page 4 lines 169-171).

REPLAY 3: Why did the authors collapse e2/e4 carriers into the e2 carrier group, considering that e2/e4 carriers are associated with increased risk in comparison to e3/e3 carriers

RESPONSE:

We thank the Referee for this comment, that allow us to clarify better this point.

We did not collapse e2/e4 carriers into the e2 carrier group: in the table 2 original version, taking a look at the numbers, we reported that e2 carriers are the sum of e2e3 + e2e4 genotypes (N=20); e3 carriers are the sum of e2e3+e3e3+e3e4 genotypes (N=332), e4 carriers are the sum of e2e4+e3e4+e4e4 genotypes (N=168).

We inserted and specified better in Material and Methods section (page 4 lines 184-186) that: ”Specifically, we performed the analyses considering e4 carriers (e2e4+e3e4+e4e4 genotypes) versus the others indicated as APOE e4 not carriers (e2e3+e3e3 genotypes)”.

Consequently, also in results (page 10 line 345; page 11 line 363) and discussion (page 13 line 451) sections, we clarified better that “APOE e4 not carriers” were associated to “Psychosis” and “Hyperactivity” endophenotypes.

In Figure 1 (page 11), these subjects were indicated as “other haplotypes”. In the note of Figure 1, we added that “other haplotypes” are referred to “APOE e4 not carriers”.

REPLAY 4: The authors do not describe how their single gene analysis was conducted in the methods section, and it is not overly clear if the results from this analysis come from logistic regression model or the CART. Did the authors include any covariates in the model such as age or sex, what model of inheritance was used? The authors should report the association of each SNP with all the NPI symptoms and the four factors developed, even if they are non-significant. This should be reported in a supplementary table if need.

RESPONSE:

We thank the Referee for these remarks, we highlighted better how we performed the analyses in this study.

1) In the Materials and Methods section (page 4 line 181-190), we specified that:

“The Classification and Regression Tree (CART) analysis using Exhaustive Chi-squared Automatic Interaction Detector (CHAID) algorithm was performed to explore the interaction between single BPSD symptoms as well as endophenotypes and all polymorphisms in APOE, MTHFR and COMT genes in the BPSD cohort [40, 41]. Specifically, we performed analyses considering e4 carriers (e2e4+e3e4+e4e4 genotypes) versus the others indicated as APOE e4 not carriers (e2e3+e3e3 genotypes). APOE allele e4 carriers, the carriers of both alleles in 677C/T MTHFR, in 1298A/C MTHFR and in Val108/158Met COMT genes polymorphisms were used as dominant model of inheritance. Moreover, the comparisons were performed incorporating “Free group” (NPI 0) along with “Low group” (NPI 1-4) versus “High group” (NPI 6-12). This is due to the evidence that “Low group” is present in our population with a frequency < 20%.”

Thus, by CART analysis using Exhaustive CHAID data mining algorithm, the results reported in Table 6 (pages 9-10) are related to single BPSD symptoms, whereas the results reported in Figure 1 (page 11) are referred to the analyses for endophenotypes. To be clearer, we added figures showing decision model trees for the genetic correlates and single BPSD symptoms that have been summarized in Table 6 (supplementary material, Figure 1S) (page 10 lines 341-342), showing the same results.

2) When we conducted the analyses on single BPSD symptoms, we did not find any “genexgene” interaction but the involvement of a “single gene”. For this reason, to avoid confusion, we eliminate “single gene” from lines 325, 334, 354, 365.

3) to avoid confusion, we modified the title in the paragraph 3.2 as following “Classification and Regression Tree (CART) Analysis for single BPSD symptoms and endophenotypes”. (page 4 line 180).

4) Concerning which model of inheritance has been used, we specified that (as already reported above):

“all analyses performed in this study were conducted considering e4 carriers (e2e4+e3e4+e4e4 genotypes) versus the others indicated as APOE e4 not carriers (e2e3+e3e3 genotypes)” (point specified in Material and Methods section, page 4 line 184-186). Thus, “APOE allele e4 carriers, but also the carriers of both alleles in 677C/T MTHFR, in 1298A/C MTHFR and in Val108/158Met COMT genes polymorphisms were used as dominant model of inheritance” (specified at page 4, lines 186-188).

5) Following the referee’s comment, we performed the analyses related to endophenotypes taking in consideration age or sex as variables in separate analyses.

We addressed this point in the Materials and Methods section, as following:

“To exclude the influence of age and gender, we performed the univariate analyses (ANOVA) related to BPSD endophenotypes, including the variables age or gender. Selecting ENDOPHENOTYPES (“Psychosis”, “Hyperactivity” “Mood” “Frontal”) as dependent variable, AGE as covariate, SEX as independent variable, we did not find any significant association in separate analyses (Tables 1S-gender and 2S-age, supplementary material). The only significant association was observed in the “Mood” endophenotype (Table 1S, F=4.7; p=0.032), data not confirmed by Exaustive CHAID data mining algorithm. Thus, given not significant analyses, age and sex were not included in the calculations.” (page 5 lines 217-224).

REPLAY 5: In the CART analysis, the authors make reference to a influenza group – I presume this is a copy/paste error.

RESPONSE:

We modified the Materials and Methods section-Statistical analyses paragraph regarding the CART analysis. The influenza group was eliminated.

REPLAY 6: How was multiple testing accounted for?

RESPONSE:

We specified that, using Exhaustive CHAID data mining algorithm:

“SPSS automatically made Bonferroni adjustment to calculate adjusted P values for the merged categories to control for Type I error rate.” (page 5 lines 215-216).

As a misprint, we substituted p correct with p adjusted (Table 6, pages 9-10). In Figure 1 (page 11), p adjusted was automatically inserted.

In the original version, we already discussed that the analyses performed were adjusted for multiple testing (page 13 lines 472-474).

Reviewer 2 Report

This is an interesting study on the clinical characterisation of AD patients. The authors have examined the association of BPSD with three AD associated genes. The results may be of interest for AD researchers. The presentation could be improved. Some comments for revision are below:

The authors do acknowledge that 90% of AD patients show neuropsychiatric disturbances. It is not clear if any additional insight on the genetic contribution for BPSD within AD patients could be gained based on the known genes for AD.

It would be useful to compare the BPSD characteristics presented in Table 1 between males and females.

In the methods section the authors mention about HWE but the results are not presented.

The single SNP analyses and the CART analysis could be presented separately.

It is not clear what is the statistical test applied for single SNP association. Age and sex effect under logistic regression model can be examined. The association results may change if adjusted for age and sex effects. All the single SNPs results with unadjusted p-values can be presented.

It would be useful to examine the NPI groups under the CART model. Predicted averages under the endophenotypes may not be useful. Effect of age and sex could also be examined under the CART model.

The authors mention about stepwise multiple logistic regression but it seems the results are not included.

Author Response

Reviewer #2: in green (gray for Referees’ #1, #2 common comments)

REPLAY 1: The authors do acknowledge that 90% of AD patients show neuropsychiatric disturbances. It is not clear if any additional insight on the genetic contribution for BPSD within AD patients could be gained based on the known genes for AD.

RESPONSE:

If we understood well the referee’s comment, we already underlined in the original version of this work, that:

1) “The susceptibility to BPSD is somewhat unclear as well as the molecular link between BPSD and AD. Within this complex aetiological interplay, genetic background has been considered one of the key players involved in predisposing patients to specific behavioural and psychological manifestations in AD [10]. Multiple genes, prevalently involved in the processes of neurotransmission/neurodevelopment, have been assessed for their putative influence on BPSD risk, whose findings have been often inconsistent (for review [10]).” (page 2 lines 62-67).

2) the genes such as APOE and COMT were involved in other previous studies both in AD and in BPSD aetiopathogenetic mechanisms (page 2 lines 68-80), whereas for MTHFR gene our study is the first to investigate its involvement in BPSD etiology , whereas previous findings found associations between this gene and late onset AD (page 2 lines 81-96).

REPLAY 2: It would be useful to compare the BPSD characteristics presented in Table 1 between males and females.

RESPONSE:

Following this comment, we compare the BPSD characteristics stratifying the sample according to the gender. The results have been reported in Table 1 (page 5) and in the results section (page 5 line 243-244).

REPLAY 3: In the methods section the authors mention about HWE but the results are not presented.

RESPONSE:

We thank the Referee for this point.

As also requested by the Referee #1, we inserted the results regarding the HWE in Table 2 (page 6) and in the results section (page 5, line 244).

Moreover, we specified that “HWE was calculated by using online calculator (http://www.husdyr.kvl.dk/htm/kc/popgen/genetik/applets/kitest.htm) for the presence of multiallelic genotypes” (page 4 lines 169-171).

REPLAY 4: The single SNP analyses and the CART analysis could be presented separately

RESPONSE:

We thank the Referee for these important comments, that allow us to clarify better how we performed the analyses in this study.

1) In the Materials and Methods section (page 4 line 181-190), we specified that:

“The Classification and Regression Tree (CART) analysis using Exhaustive Chi-squared Automatic Interaction Detector (CHAID) algorithm was performed to explore the interaction between single BPSD symptoms as well as endophenotypes and all polymorphisms in APOE, MTHFR and COMT genes in the BPSD cohort [40, 41]. Specifically, we performed analyses considering e4 carriers (e2e4+e3e4+e4e4 genotypes) versus the others indicated as APOE e4 not carriers (e2e3+e3e3 genotypes). APOE allele e4 carriers, the carriers of both alleles in 677C/T MTHFR, in 1298A/C MTHFR and in Val108/158Met COMT genes polymorphisms were used as dominant model of inheritance. Moreover, the comparisons were performed incorporating “Free group” (NPI 0) along with “Low group” (NPI 1-4) versus “High group” (NPI 6-12). This is due to the evidence that “Low group” is present in our population with a frequency < 20%.”

Thus, by CART analysis using Exhaustive CHAID data mining algorithm, the results reported in Table 6 are related to single BPSD symptoms, whereas the results reported in Figure 1 are referred to the analyses for endophenotypes. To be clearer, we added figures showing decision model trees for the genetic correlates and single BPSD symptoms that have been summarized in Table 6 (supplementary material, Figure 1S) (page 10 lines 341-342), showing the same results.

2) When we conducted the analyses on single BPSD symptoms, we did not find any “genexgene” interaction but the involvement of a “single gene”. For this reason, to avoid confusion, we eliminate “single gene” from lines 325, 334, 354, 365.

3) to avoid confusion, we modified the title in the paragraph 3.2 as following “Classification and Regression Tree (CART) Analysis for single BPSD symptoms and endophenotypes”. (page 4 line 180).

REPLAY 5: It is not clear what is the statistical test applied for single SNP association. Age and sex effect under logistic regression model can be examined. The association results may change if adjusted for age and sex effects. All the single SNPs results with unadjusted p-values can be presented.

RESPONSE:

1) The statement “single gene” has probably confused the analyses performed.

As above reported, when we conducted the analyses on single BPSD symptoms, we did not find any “genexgene” interaction but the involvement of a “single gene”. For this reason, to avoid confusion, we eliminate “single gene” from lines 325, 334, 354, 365. CART analysis using Exhaustive CHAID data mining algorithm was conducted both according to single BPSD symptoms (Table 6, pages 9-10; see also Figure 1S), and to endophenotypes clusterization (Figure 1, page 11).

2) Following the Referee’s comment, we performed the analyses related to endophenotypes taking in consideration age or sex as variables in separate analyses.

We addressed this point in the Materials and Methods section, as following:

“To exclude the influence of age and gender, we performed the univariate analyses (ANOVA) related to BPSD endophenotypes, including the variables age or gender. Selecting ENDOPHENOTYPES (“Psychosis”, “Hyperactivity” “Mood” “Frontal”) as dependent variable, AGE as covariate, SEX as independent variable, we did not find any significant association in separate analyses (Tables 1S-gender and 2S-age, supplementary material). The only significant association was observed in the “Mood” endophenotype (Table 1S, F=4.7; p=0.032), data not confirmed by Exhaustive CHAID data mining algorithm. Thus, given not significant analyses, age and sex were not included in the calculations.” (page 5 lines 217-224).

REPLAY 6: It would be useful to examine the NPI groups under the CART model. Predicted averages under the endophenotypes may not be useful. Effect of age and sex could also be examined under the CART model.

RESPONSE:

As above reported, CART analysis using Exhaustive CHAID data mining algorithm was conducted both according to single BPSD symptoms (Table 6, pages 9-10; see also Figure 1S), and to endophenotypes clusterization (Figure 1, page 11).

As already specified, given not significant analyses, age and sex were not included in the calculations.

REPLAY 7: The authors mention about stepwise multiple logistic regression but it seems the results are not included.

RESPONSE:

We thank the Referee for this replay.

We added this comment in the Materials and Methods section (page 5 line 229), whereas the results concerning the stepwise logistic regression in results section (page 9 line 320-321) and added Tables 4S and 5S as supplementary material. The data evidenced that the stepwise multiple logistic regression validated the classification trees.  

Reviewer 3 Report

Summary

Authors defined symptoms of BPSD using the Neuropsychiatric Inventory (NPI) scale in an Italian cohort of Alzheimer's disease (AD) patients. Principle Component Analysis was used to reduce NPI symptoms into specific categories of BPSD symptoms in AD. Author's identified the following categories in the AD sample: “Psychosis”, “Hyperactivity” “Mood” and “Frontal” (e.g., disinhibition and euphoria). CART, a clustering algoritim that includes causality analysis, was applied to the genetic and BPSD data, allowing author to infer genetic causality behind the BPSD symptoms and previously identified categories. The APOE genotype is associated with more severe “Psychosis”. Authors also find that the APOE and MTHFR interact to predict psychosis severity, as well as hyperactive as a whole category. The APOE 4 allele was associated with higher severity in appetite/eating abnormalities, from the “Hyperactivity” category. No genetic determinant for last two categories (clusterisations) in the AD sample were identified (i.e. Mood and Frontal).  The large sample size AD patients (n=362) is a strength and ensures that the clustering approach applied to the dataset is suitable. The discussion is well organised and provides a good interpretation of the results in the context of existing research. The article fits with the aims of the Journal of Personalised Medicine, as it draws on the genetic profiling of AD patients to infer what BPSD symptoms are expected from a specific AD patient group.

Specific comments

Abstract

- Sentence 22-24: Please rephrase. See example:

The definition of the clinical distribution of BPSD symptoms, and the extent to which symptoms are genetically determined, is still open to debate. Moreover, genetic factors that underlie BPSD symptoms still need to be identified”

-  Please clarify what is meant by “single” in the following sentence

To characterize our Italian AD cohort according to BPSD symptoms (both single and endophenotypes)”

- Line 28 typo: “seriuos”

- Line 43 typo: “ needed.

- Line 55-57: please rephrase

- The statement on line 58-61 requires references.

Methods

- Line 144: Spectrophotometric quantification - please provide reference

- Was medication (e.g., AchEIs, Neuroleptics, Antidepressants, Benzodiazepines/hypnotics) adjusted for when investigated the genetics correlates clusterisation and single symptoms? If medication influences the severity of 12 NPI symptoms, is in turn will confound the genetic correlation findings. If medication was not considering in the analysis, please state this in the limitations.

- No reference to whether the data was normally distrubted. If this is not required for the CART clustering approach for example, authors should state that in the methods.

- Please include the result for the “Hosmer-Lemeshow goodness of fit test’’ (in supplementary materials if possible) to confirm the suitability of approach to the data.

Results

- Table 1: Please provide information on the abbreviations in the footer notes.

- Figure 1 is hard to interpret due to small font size and low resolution of the image

- Please include plots to show that the data was normality disturbed, as there is currently no reference to this, nor is there any reference to data transforms (e.g., log)

- In Table 2 (See picture), 194 e4 alleles are stated. However, authors also include the more important information ‘’168 e4 allele carriers’’. What is the rationale for include the number of single ‘ alleles’ for example e4 alleles)? 

- Line 250 typo: meccanisms

- Line 259 typo: rish

- Line 261 typo: homozigotes

- In APOE, MTHFR and COMT genetic correlates in endophenotype clusterization and in single BPSD symptoms section, authors state that carriers of a gene/allele are at high risk to develop more severe symptoms. These results are not based on longitudinal tracking of the AD patients. If authors are assuming causality based on the CART method applied (i.e. causal analysis), please state that in the methods.

Discussion

- Authors should include a reference to the prevalence of each gene studied and how prevalence guides the importance of one finding over the other. For example, e3e4 genotype is four times more common than e4e4 genotype. Thus, it may be more important to understand e3e4 contributions to BPSD symptoms given that they are more representative of the preclinical and clinical AD population.

- Line 363 includes a typo.

The statement on line 329-330 requires referencing.

Author Response

Reviewer #3: in heavenly

Authors defined symptoms of BPSD using the Neuropsychiatric Inventory (NPI) scale in an Italian cohort of Alzheimer's disease (AD) patients. Principle Component Analysis was used to reduce NPI symptoms into specific categories of BPSD symptoms in AD. Author's identified the following categories in the AD sample: “Psychosis”, “Hyperactivity” “Mood” and “Frontal” (e.g., disinhibition and euphoria). CART, a clustering algorithm that includes causality analysis, was applied to the genetic and BPSD data, allowing author to infer genetic causality behind the BPSD symptoms and previously identified categories. The APOE genotype is associated with more severe “Psychosis”. Authors also find that the APOE and MTHFR interact to predict psychosis severity, as well as hyperactive as a whole category. The APOE 4 allele was associated with higher severity in appetite/eating abnormalities, from the “Hyperactivity” category. No genetic determinant for last two categories (clusterizations) in the AD sample were identified (i.e. Mood and Frontal).  The large sample size AD patients (n=362) is a strength and ensures that the clustering approach applied to the dataset is suitable. The discussion is well organised and provides a good interpretation of the results in the context of existing research. The article fits with the aims of the Journal of Personalised Medicine, as it draws on the genetic profiling of AD patients to infer what BPSD symptoms are expected from a specific AD patient group.

REPLAY 1. Abstract

- Sentence 22-24: Please rephrase. See example:

“The definition of the clinical distribution of BPSD symptoms, and the extent to which symptoms are genetically determined, is still open to debate. Moreover, genetic factors that underlie BPSD symptoms still need to be identified”

RESPONSE:

We thank the Referee, and we corrected as requested (lines 23-25).

REPLAY 2 Abstract

Please clarify what is meant by “single” in the following sentence

“To characterize our Italian AD cohort according to BPSD symptoms (both single and endophenotypes)”

RESPONSE:

We clarified as following:

“To characterize our Italian AD cohort according to specific BPSD symptoms as well as to endophenotypes” (lines 26-27).

REPLAY 3 Abstract

- Line 28 typo: “seriuos”

- Line 43 typo: “ needed.

- Line 55-57: please rephrase

RESPONSE:

We corrected each typing errors.

The sentence was rephrased as following:

“However, the debate about the definition of an appropriate clusteritation in dementia is still ongoing [3]. More importantly, the studies available to date are characterized by a high heterogenity, also because a certain number of symptoms (i.e. apathy, sleep disorders, eating disturbances) are not adeguately grouped [3]” (page 2 lines 58-61).

REPLAY 4. The statement on line 58-61 requires references.

RESPONSE:

We added the reference “Flirski, M.; Sobow, T.; Kloszewska, I. Behavioural genetics of Alzheimer's disease: a comprehensive review. Arch. Med. Sci. 2011, 7, 195-210.” [10] (page 2 line 65).

REPLAY 5. Methods

- Line 144: Spectrophotometric quantification - please provide reference

RESPONSE:

We added the following sentence: “The obtained gDNA was quantified by spectrophotometric quantification using NanoDrop microvolume sample retention system (Thermo Fisher Scientific, Waltham, MA)” (pages 3-4 lines 152-154) and the relative reference: “[37]: Desjardins P, Conklin D. NanoDrop microvolume quantitation of nucleic acids. J Vis Exp. 2010;(45):2565. Published 2010 Nov 22. doi:10.3791/2565”. (page 4 line 154).

REPLAY 6. Methods

Was medication (e.g., AchEIs, Neuroleptics, Antidepressants, Benzodiazepines/hypnotics) adjusted for when investigated the genetics correlates clusterization and single symptoms? If medication influences the severity of 12 NPI symptoms, is in turn will confound the genetic correlation findings. If medication was not considering in the analysis, please state this in the limitations.

RESPONSE:

We thank the Referee for this point.

The variables “Medications” as well as “Comorbidity” were not considered in our analyses. Thus, we stated this in the Materials and Methods section (Page 5 lines 224-225), and in limitation section inserted in the discussion (pages 13-14, lines 474-480), we added that:

“Medications as well as comorbidity are clinical information extremely different in a BPSD population, also including our cohort (Table 1). To include “The only variable not included in our analyses was “Medications”. Medications as well as comorbidity are clinical information extremely heterogeneous in a BPSD population, including our cohort (Table 1). To include “Medications” or “Comorbidity” as variables in the statistical analyses would result to be overly complex, especially in a sample of sample sizes like ours. Thus, we suggest that future studies should consider these variables, but starting from homogeneous groups for medications and for comorbidity, on which then to investigate the genetic correlates.”

REPLAY 7. Methods

- No reference to whether the data was normally distributed. If this is not required for the CART clustering approach for example, authors should state that in the methods.

RESPONSE:

We thank the Referee for this right comment.

In the Materials and Methods section, we clarified and added this point:

“Exhaustive CHAID data mining algorithm is nonparametric procedure that make no assumptions of the underlying data. In CHAID analysis, nominal, ordinal, and continuous data can be used, where continuous predictors are split into categories with approximately equal number of observations. CHAID creates all possible cross tabulations for each categorical predictor until the best outcome is achieved and no further splitting can be performed [42]. In the CHAID technique, we can visually see the relationships between the split variables and the associated related factor within the tree. The development of the decision, or classification tree, starts with identifying the target variable or dependent variable; which would be considered the root. CHAID analysis splits the target into two or more categories that are called the initial, or parent nodes, and then the nodes are split using statistical algorithms into child nodes.” (page 4 lines 194-203).

REPLAY 8. Methods

- Please include the result for the “Hosmer-Lemeshow goodness of fit test’’ (in supplementary materials if possible) to confirm the suitability of approach to the data.

RESPONSE:

In the supplementary material (Table 3S), we showed the results about “Hosmer-Lemeshow goodness of fit test’’. This point was added in the Material and Methods section (page 5 lines 226-228) and also in the results section (page 9 lines 318-319). The Hosmer-Lemeshow goodness of fit test confirmed the suitability of the trees.

REPLAY 9. Results

- Table 1: Please provide information on the abbreviations in the footer notes.

- Figure 1 is hard to interpret due to small font size and low resolution of the image

RESPONSE:

We provided information requested in Table 1 (page 6). Figure 1 was better defined (Figure 1_REV).

REPLAY 10. Results

- Please include plots to show that the data was normality disturbed, as there is currently no reference to this, nor is there any reference to data transforms (e.g., log)

RESPONSE:

Because the analyses were conducted by using decision tree models, it is not needed to have normalized data, thus we did not perform transformed analyses. This point was added in the Material and Methods section (page 5 line 213-214). 

REPLAY 11. Results

- In Table 2 (See picture), 194 e4 alleles are stated. However, authors also include the more important information ‘’168 e4 allele carriers’’. What is the rationale for include the number of single ‘ alleles’ for example e4 alleles)?

RESPONSE:

There is not a rationale to include the number of single alleles for APOE gene. Table 2 (page 6) shows the allele and genotype distributions of all polymorphisms investigated in this study in APOE, COMT and MTHFR genes, inside our AD cohort.

We explained better in Materials and Methods section (page 4 lines 184-186) that:

“Specifically, we performed the analyses considering e4 carriers (e2e4+e3e4+e4e4 genotypes) versus the others indicated as APOE e4 not carriers (e2e3+e3e3 genotypes)”. Consequently, also in results (page 10 line 345; page 11 line 363) and discussion (page 13 line 451) sections, we clarified better that “APOE e4 not carriers” were associated to “Psychosis” and “Hyperactivity” endophenotypes.

In Figure 1 (page 11), these subjects were indicated as “other haplotypes”. In the note of Figure 1, we added that “other haplotypes” are referred to “APOE e4 not carriers”.

REPLAY 12. Results

- Line 250 typo: meccanisms

- Line 259 typo: rish

- Line 261 typo: homozigotes

RESPONSE:

We thank the Referee to check these typing errors.

We corrected each of them: line 318, 331, 346- 347 respectively.

REPLAY 13. Results

In APOE, MTHFR and COMT genetic correlates in endophenotype clusterization and in single BPSD symptoms section, authors state that carriers of a gene/allele are at high risk to develop more severe symptoms. These results are not based on longitudinal tracking of the AD patients. If authors are assuming causality based on the CART method applied (i.e. causal analysis), please state that in the methods.

RESPONSE:

We thank the Referee for this specific point. Indeed, in the original version, we discussed this evidencing that “longitudinal studies are required to achieve significant insights into the evolution of BPSD during the course of disease [3].“ (page 12 lines 421-422).

We added in the Materials and Methods section (page 4 lines 190-193) that:

“As this study has not a longitudinal design, we assumed causality in our CART model (https://www.epa.gov/caddis-vol4/caddis-volume-4-data-analysis-classification-and-regression-tree-cart-analysis)”.

REPLAY 14. Discussion

- Authors should include a reference to the prevalence of each gene studied and how prevalence guides the importance of one finding over the other. For example, e3e4 genotype is four times more common than e4e4 genotype. Thus, it may be more important to understand e3e4 contributions to BPSD symptoms given that they are more representative of the preclinical and clinical AD population.

RESPONSE:

We checked the prevalence of each polymorphisms in APOE, MTHFR and COMT genes in https://www.alzforum.org/ .

We added in the paragraph “3. APOE, MTHFR and COMT genetic correlates in endophenotype clusterization and in single BPSD symptoms” that:

“All polymorphisms investigated in this study within APOE, MTHFR and COMT genes showed a prevalence comparable to that reported in https://www.alzforum.org/, considering Caucasian Populations (Table 2). Globally, the prevalence of ε2, ε3 and ε4 alleles for APOE gene was estimated to be 8, 78 and 14 %, respectively; for rs1801133 MTHFR gene to be 39% (allele T) and 61% (allele C); for rs1801131 MTHFR gene to be 34% (allele C) and 66% (allele A); for Val108/158Met COMT gene to be 49% (allele A) and 51% (allele A).“ (page 9 lines 309-314).

Moreover, to be clearer, we explained better in Materials and Methods section (page 4 lines 184-186) that:

“Specifically, we performed the analyses considering e4 carriers (e2e4+e3e4+e4e4 genotypes) versus the others indicated as APOE e4 not carriers (e2e3+e3e3 genotypes)”. Consequently, also in results (page 10 line 345; page 11 line 363) and discussion (page 13 line 451) sections, we clarified better that “APOE e4 not carriers” were associated to “Psychosis” and “Hyperactivity” endophenotypes.

In Figure 1 (page 11), these subjects were indicated as “other haplotypes”. In the note of Figure 1, we added that “other haplotypes” are referred to “APOE e4 not carriers”.

REPLAY 14. Discussion

Line 363 includes a typo.

RESPONSE:

We corrected the typo indicated (page 13 line 439).

REPLAY 14. Discussion

The statement on line 329-330 requires referencing.

RESPONSE:

We added the reference requested: “[43]: Deardorff, W. J.; Grossberg, G. T. Behavioral and psychological symptoms in Alzheimer's dementia and vascular dementia. Handb. Clin. Neurol. 2019, 165, 5-32.” (page 12 line 405).
